# Continuous Parametric Optical Flow

**Jianqin Luo**∗    **Zhexiong Wan**∗    **Yuxin Mao**    **Bo Li**    **Yuchao Dai**†

Northwestern Polytechnical University, Xi'an, China
Shaanxi Key Laboratory of Information Acquisition and Processing

## Abstract

In this paper, we present *continuous parametric optical flow*, a parametric representation of dense and continuous motion over *arbitrary time interval*. In contrast to existing discrete-time representations (*i.e.* flow in between consecutive frames), this new representation transforms the frame-to-frame pixel correspondences to dense continuous flow. In particular, we present a temporal-parametric model that employs B-splines to fit point trajectories using a limited number of frames. To further improve the stability and robustness of the trajectories, we also add an encoder with a neural ordinary differential equation (NODE) to represent features associated with specific times. We also contribute a synthetic dataset and introduce two evaluation perspectives to measure the accuracy and robustness of continuous flow estimation. Benefiting from the combination of explicit parametric modeling and implicit feature optimization, our model focuses on motion continuity and outperforms the flow-based and point-tracking approaches for fitting long-term and variable sequences.

## 1   Introduction

Motion estimation, which characterizes the movement of targets over time, is crucial for video understanding tasks like keypoint tracking [1] or video interpolation [2]. It also provides accurate priors for 3D tasks such as non-rigid structure-from-motion [3, 4] and simultaneous localization and mapping [5]. Fine-grained motion understanding is necessary to achieve pixel-level correspondence, which is usually studied as an optical flow estimation problem. Some traditional methods [6, 7] based on variational optimization or recent deep methods focusing on the Tracking-Any-Point (TAP) task [8, 9] both attempt to estimate long-range flow for tracking points across multi-frames.

While these approaches have made significant progress in modeling correspondences, there still exist two obstacles to explaining and predicting practical motion, as shown in Fig. 1. First, the point trajectories generated by these approaches only reflect the possible locations at certain moments instead of real motion trends because the frame-to-frame correspondence is temporally discrete and lacks enough inter-frame information to describe continuous point motion. Second, optical flow-based methods could achieve dense but short-term point mapping while recent point trackers tend to strengthen long-term tracking for sparse points. Neither can efficiently realize dense and long-term tracking simultaneously.

In this paper, we aim to estimate the dense and continuous pixel correspondence with the continuous flow that provides the moving trajectory from reference time to arbitrary target times within the whole

---

∗These authors contributed equally to this work.
†Corresponding author (daiyuchao@nwpu.edu.cn).
‡Project page: https://npucvr.github.io/CPFlow .

37th Conference on Neural Information Processing Systems (NeurIPS 2023).

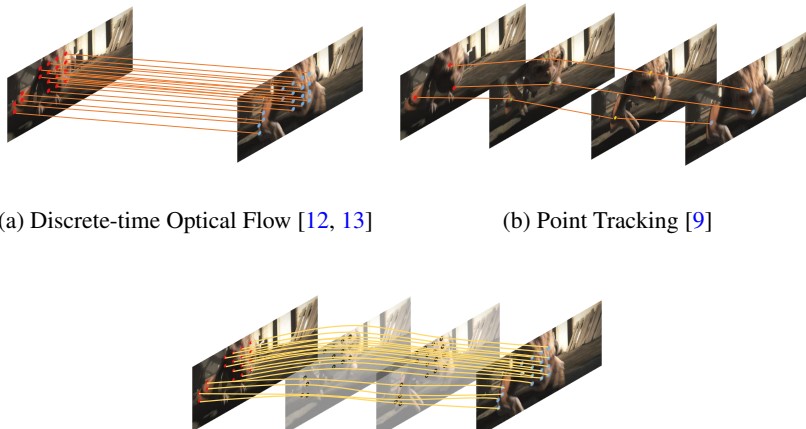

(a) Discrete-time Optical Flow [12, 13]          (b) Point Tracking [9]

(c) Continuous Optical Flow (Ours)

Figure 1: Comparison between continuous optical flow with spatially dense and temporally continuous attributes with discrete-time flow or pixel tracking.

video. We utilize a learnable parametric model to explicitly describe the continuous flow trajectory, which avoids some inconsistent and abrupt displacements compared to the implicit counterparts directly predicting the final result. Motivated by [10], we embed the continuous property into our end-to-end optimization pipeline to enhance continuity in feature space. By combining the powerful representational ability and explicit motion pattern, our method is more appropriate for solving continuous problems. To evaluate the continuous flow estimation, we introduce two evaluation perspectives, including continuous motion with invisible frames at specific timescales and robustness validation at variable timescales. We also use Kubric [11] to create a synthetic dataset with dense and long trajectories for experiments.

Our main contributions are summarized as:

- We present continuous parametric optical flow, a temporal parametric representation of dense and continuous motion over arbitrary time intervals.
- We utilize Neural ODE and ConvGRU to aggregate the spatial-temporal information and enhance the continuity of features among target times.
- We create a large-scale synthetic dataset with dense and long-term trajectories and propose a two-stage evaluation pipeline to measure the accuracy and robustness of continuous flow estimation.

## 2 Related Work

### 2.1 Optical Flow Estimation

Optical flow estimation is a fundamental task in computer vision, and existing optimization-based methods [14, 15, 16, 17] have made explorations in challenging situations such as large motions [18, 19] and occlusions[20, 21]. While early learning-based methods take an encoder-decoder [22] or coarse-to-fine [23, 24] structure, recent methods, such as RAFT [12] and GMA [25], usually take an iterative update pipeline. These methods usually take procedures of supervised training on simulated data [25, 13] or unsupervised training on real data [26, 27].

Unlike the above common two-frame input, some methods [26, 28] use multi-frames to aggregate additional flows to tackle the occlusion. MFSF [6] aims to establish the long-range correspondence of multi-frame flow by a linear combination of trajectory basis. SlowFlow [29] links adjacent flows to longer trajectories by constraining brightness, smoothness, and visibility. Gehrig et al. [30] proposes to use event data to build motion trajectories for fast and non-linear motion in a short period.

## 2.2 Feature and Point Tracking

Among the self-supervised feature tracking methods, TimeCycle [31] uses cycle consistency as the primary constraint for patch-level extraction and matching. CRW [32] designs a special palindrome arrangement to introduce cycle loss. VFS [33] introduces contrastive learning to optimize frame-level matching. Instead of implicit exploration in unlabeled data, supervised methods [34, 35] rely on ground truth correspondence to learn a general feature representation. Although these methods perform well on general tasks like keypoint tracking or semi-supervised video segmentation, pixel-level matching remains unsolved because image or patch embedding seems insufficient for specific point refinement.

For long-term point tracking, the key problem is how to match the spatial-temporal features of the long-range correspondences. PIPs [9] proposes to extract query point features for pixel tracking and occlusion estimation with iterative updates, motivated by Particle Video [36]. TAP-Net [8] uses a flow-based method to assist annotators to track sparse points selected by random sampling and generate point trajectories with longer duration.

## 2.3 Parametric Motion Hypothesis

Rigid motion modeling was a fashion for early work before various modern learning techniques. As for spatial modeling, some representative methods [37, 38] use robust parametric motion like affine transformation or based on over-parameterization [39, 40] to estimate flow. Yang and Li [41] propose a piecewise, 8-Dof homography model to fuse multiple possible parametric motions. Besides, layered models [42, 43] also perform competitively with a few overlapping motion layers that balance the need for smoothness and boundary separation.

For temporal consistency, the linear combination of motion basis [6, 44, 45] is a common method for trajectory fitting in the time axis. These pre-defined bases like DCT [6] or Lagrange [44] can convert the continuous time domain into the discrete analysis. With different weight assignments for basis groups, this modeling approach is convenient to implement for deep paradigm [45]. However, limited by the sampling range, these models are not flexible for video clips with different frame numbers. Another alternative is to perform temporally continuous curve fitting, including linear [46], quadratic [47], or cubic [48] curve hypotheses with polynomial modeling.

In addition, complex motion trajectories can be represented by adaptive curves with a few control points like B-splines [49] or Bezier [30]. Although most of these parametric hypotheses were first introduced in previous optimization methods, modern methods can still take a deep network as the process of motion parameterization by regressing a set of coefficients or control points.

# 3 Parametric Modeling

## 3.1 Preliminaries of Continuous Flow

To describe the continuous flow that expresses the motion from the reference time to any target time, we define a triplet $(t, u, v)$ as a query of flow trajectory decoupled into two components along horizontal and vertical direction for corresponding pixel $(x, y)$, $t$ is sampled from normalized time $T \in [0, 1]$. For brevity, the reference time is zero. Instead of directly fitting the pixel position, we choose to estimate the pixel displacements and the start point of flow trajectory $(u, v) = (0, 0)$. Note that although we can only use finite labels to evaluate our model, we introduce two evaluation perspectives to measure the performance under diverse situations.

**Stage 1: Blind Estimation.** Given a specific number of frames at sampled timestamps with a fixed timescale as input, the model should predict the target flow trajectory at unsampled times.

**Stage 2: Temporal Adaptation.** The pre-trained model on a fixed training range should adapt to diverse sequences with different timescales.

## 3.2 Parametric Curve for Continuous Flow Modeling

Given $N$ control points $\{P_i = (x_i, y_i)\}^N \in \mathbb{R}^{N \times 2}$ and basis functions $B_{i,k}(t)$ with degree $k$, the point displacement trajectory $F(t)$ is formulated as:

$$F(t) = \sum_{i=0}^{N-1} B_{i,k}(t) P_i. \tag{1}$$

For any specific time $t$, the basis functions return a group of weights to determine the contributions for the curve shape of each control point. The parametric curve we employ is a cubic B-spline formed by a few learnable control points and spline basis functions. In our work, the coordinates of $N$ control points serve as model outputs, whereas the normalized knots $\{t_i\}$ are manually specified hyperparameters for local optimization in the temporal range. The detailed derivation of basis functions is referred to Cox-de Boor recursion [50, 51]:

$$B_{i,0}(t) = \begin{cases} 1 & t_i \leq t < t_{i+1} \\ 0 & \text{otherwise} \end{cases}, \tag{2}$$

$$B_{i,k}(t) = \frac{t - t_i}{t_{i+k} - t_i} B_{i,k-1}(t) + \frac{t_{i+k+1} - t}{t_{i+k+1} - t_{i+1}} B_{i+1,k-1}(t). \tag{3}$$

We assign each time to a single partition to ensure the sum of the weight coefficients of all control points is equal to 1. Knots can be repeated to promote the curve from open to close. In our experiment, a special shape like the clamp seems easier to reach convergence. Benefiting from the potential advantages of parameter design, the cubic B-spline could achieve the combination of global relaxation and fine-tuning on local segments.

The parametric curve is optimized independently for each pixel to build a complete displacement flow field. As a result, our model directly predicts a 3D tensor with the shape of $2N \times H \times W$ for control points in all trajectories.

## 4 Optimization

The optimization process consists of four components: multi-frame sampling, implicit continuity enhancement, multi-time correlation pyramids, and iterative control point decoder. According to the order of the input video clip with $L$ frames, the corresponding timestamps are normalized to $[0, 1]$. Our framework aims to regress a set of control points with their 2D coordinates for each pixel and to compute the final continuous flow trajectories by Eq. (1).

### 4.1 Multi-frame Sampling

For a given video clip with $L$ frames, the model input $M$ frames are first sampled from the entire sequence as visible moments in the continuous time domain while the remaining $L - M$ frames are assumed to be invisible. We use the ground-truth displacement fields sampled from all valid moments corresponding to the frames as the supervision of continuous flow trajectories for model training.

During training, we first select the start frame with a sufficient number of visible points from the given video clip. Then we randomly sample a subsequence clip with $M$ frames with different intervals as model inputs to enhance the generation ability to different motion magnitudes. Notably, this selection process keeps the start and end frames and only randomly selects the intermediate $M - 2$ frames based on brightness differences.

### 4.2 Implicit Continuity Enhancement

As shown in Fig. 2, we first extract $M$ basic feature maps using a shared basic encoder for each of the input $M$ frames. Similar to RAFT [12], we also feed the reference frame (*i.e.* the start frame with $t = 0$) into the context encoder for extracting another context feature to assist subsequent motion estimation. Unlike PIPs [9], we introduce ODE-ConvGRU proposed in [2] for spatio-temporal feature learning. This module consists of the Neural ODE for temporal continuity enhancement in feature space and the Conv-GRU for spatio-temporal information aggregation. The Neural ODE formulated

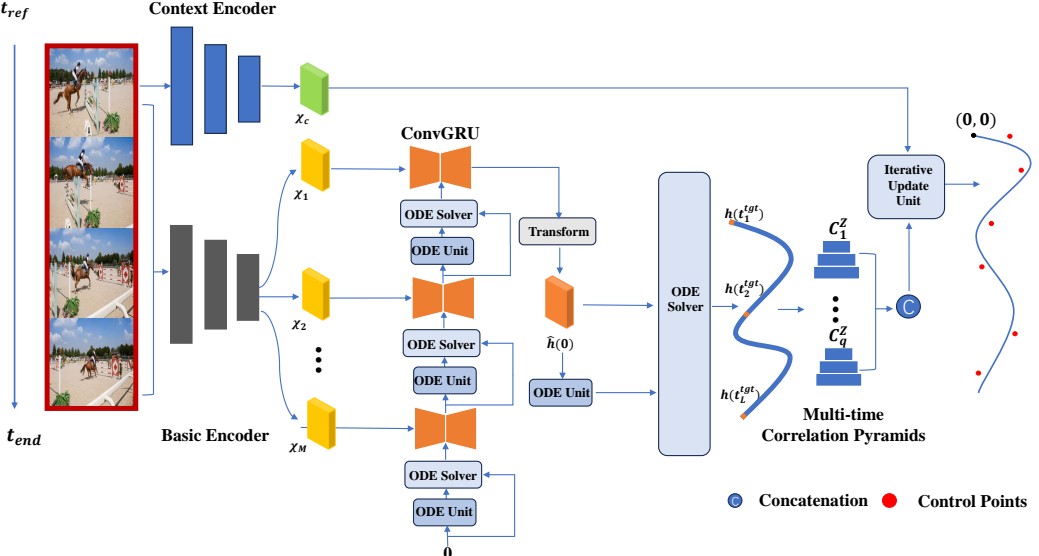

Figure 2: Pipeline of our model, combined with continuous temporal-spatial feature and explicit motion trajectory curves. ODE-ConvGRU enhances spatial features with continuous temporal information and generates powerful representations at arbitrary target moments.

by Eq. (4) includes a derivative estimator $f$ as an independent ODE unit and an ODE solver, in which the parameters involve the initial value $h(t_0)$, the current time $t$ and input state $h_{in}(t)$. The former uses a learnable fully-connected network $f$ to fit the feature differentiation at time $t$, and the latter integrates to recover the hidden state $h(t)$ at $t$ started from $t_0$.

$$h(t) = \text{ODE}(f, t, h_{in}(t)) \quad \begin{cases} \dfrac{dh_{in}(t)}{dt} = f(h_{in}(t), t), \\ h(t) = h_{in}(t_0) + \displaystyle\int_{t_0}^{t} f(h_{in}(t), \tau)d\tau. \end{cases} \tag{4}$$

The timestamps corresponding to the input $M$ frames serve as **source times** $\{t_j^{src}\}^M$ to feed into the ODE-ConvGRU module and output features at **target times** $\{t_j^{tgt}\}^L$ for correlation construction. The Neural ODE models the features of the entire $L$-frame video clip including invisible frames, so it also outputs $L$ target times.

Our ODE-ConvGRU module consists of three steps. First and foremost, we compute the initial hidden states $\overline{h}(t_j^{src})$ at source times $t_j^{src}$ by integrating the derivative of the previous hidden state at $t_{j-1}^{src}$ solved by network $f_\theta$:

$$\overline{h}(t_j^{src}) = \text{ODE}(f_\theta, \{t_j^{src}\}, \overline{h}(t_{j-1}^{src})), \tag{5}$$

where this process starts with $\overline{h}(t_0^{src}) = 0$, *i.e.*, an zero feature map. Note that the computation process is backward, which means the original temporal order needs to be reversed.

Besides, given feature $\chi$ extracted by the basic encoder, we update the initial hidden state $\overline{h}(t_j^{src})$ at source time $t_j^{src}$ by aggregating them to generate refined hidden state $\tilde{h}(t_j^{src})$ with ConvGRU:

$$\tilde{h}(t_j^{src}) = \text{ConvGRU}(\overline{h}(t_j^{src}), \chi_{t_j^{src}}). \tag{6}$$

Finally, the last output hidden state $\tilde{h}(0)$ from ConvGRU aggregates valid information from all source moments, and is transformed into a novel initial value $\hat{h}(0)$ by a small network $g_\omega$. The derivative is computed by another ODE unit with network $f_\phi$ and evolves into the final hidden state outputs $h(t_j^{tgt})$ at target time $t_j^{tgt}$:

$$\{h(t_j^{tgt})\}^L = \text{ODE}(f_\phi, \{t_j^{tgt}\}^L, \hat{h}(0)). \tag{7}$$

### 4.3 Multi-time Correlation Pyramid

To achieve fine-grained matching, we compute the spatial matching feature similarity between different target times. Due to the powerful spatio-temporal representation of ODE-ConvGRU, we can obtain an arbitrary number of feature pairs for continuous flow estimation. Thus we can perform the correlation operation between the reference state $h(t_0^{tgt})$ and the rest of hidden states $\{(h(t_j^{tgt})\}_1^L$ at target time to construct $L-1$ cost volumes.

However, due to limited GPU memory, we still need to introduce sampling to reduce computational complexity while leveraging the continuous representation capability. We make a trade-off by random sampling $Q$ ($Q \leq (L-1)$) hidden states to construct $Q$ multi-time correlations $C = \{C_q\}^Q$. We then perform the average pooling operation and obtain the correlation pyramids $C_q = \{C_q^z\}^Z$ ($Z$ is the pooling level) for each correlation separately, which covers matching similarities at multiple resolutions for subsequent local lookup.

### 4.4 Iterative Lookup and Update

Inspired by RAFT [12], we also adopt the lookup operation to search the local matching patches from multi-time correlations. Subsequently, we concatenate the patches with different moments and different pooling levels as a single tensor and feed it into the decoder to regress the residual of control points and compute the continuous optical flow. This procedure of local lookup and flow regression is iteratively executed by adjusting the local search starting point according to the updated control points.

For $N$ control points $\{P_i\}^N$, the update unit outputs $N$ translations $\{\Delta P_i\}^N$ and updates iteratively:

$$P_i^{s+1} = P_i^s + \Delta P_i^s, \tag{8}$$

where $i$ is the index of the control point and $s$ is the iteration number for lookup and update.

### 4.5 Supervision

As there are only discrete point displacement ground truths in the dataset, we can only use them to discretely supervise the predicted continuous optical flow of our model. Assuming that for each point there are $N_{gt}$ displacement ground truths at different times, we calculate the point displacements on the predicted continuous trajectory with the corresponding times $\{t_w^{sup}\}^{N_{gt}}$ according to Eq. (1).

The loss function measures the average displacement error at $N_{gt}$ different times and all visible pixels with a weighted sum of all iterations:

$$L = \sum_{i=1}^{S} \gamma^{S-i} \frac{1}{N_{gt}} \sum_{j=1}^{N_{gt}} ||F(t_j^{sup}) - D_{gt}(t_j^{sup})||_1, \tag{9}$$

where the iteration times is $S$ and weight is $\gamma$. $F(t_w^{sup})$ is the point displacements at time $t_w^{sup}$ on the predicted continuous trajectory, $D_{gt}(t_j^{sup})$ is the the corresponding point displacement ground truth. Note that the starting point of the continuous trajectory is 0, so we omit it.

## 5 Experiments

This section consists of four associated parts. We first use Kubric [11] to simulate a relatively dense and longer pixel-tracking dataset for our model training and as a basic benchmark for evaluation. Second, to establish valid baselines, we introduce two approaches [9, 12] to provide direct estimation for sampling moments and parametric assumption to simulate inter-frame motion. Third, we set error metrics for quantitative comparison and show the results on synthetic scenes and real-world scenes. Finally, ablation experiments are implemented for module validity tests.

We train our model on four RTX 3090 GPUs with a batch size of 8. At the training stage, we use the resolution of 256×256 and randomly sample 20480 point trajectories as sparse supervisions for better performance. We train for 30,000 steps because more iterations may result in degradation on diverse timescales with a learning rate of 2e-5 and a one-cycle schedule [52]. We repeat multiple times and report median results. We set the numbers of control points $N = 6$ while the degree $k = 3$,

and the sampling number of frames $M = 4$. The number $Q$ of correlation is 3, and the pooling level $Z$ is 3. In the supervision process, the $N_{gt} = 8$ and $\gamma = 0.8$. For temporal augmentation, the length of video clips $L$ is variable but at least 8 frames.

## 5.1 Dataset

To maintain the same settings as [8, 9], we utilize the Kubric simulator to generate two subsets: query-stride and query-first in our experiment. The video samples in both the two subsets are with 24 frames at 256×256 resolution. Points trajectory is sampled from the whole spatial-temporal area in the query-stride version but from the single frame in the query-first version. Query-stride is applied in the training phase for less structured constraints and query-first can be used for dense evaluation.

## 5.2 Metrics

We use two metrics to measure the accuracy at specific moments and the temporal trajectory smoothness of continuous flow estimation. From a spatial perspective, the average displacement error (ADE) proposed in [9] is applied and divided into visible and occluded parts. To reflect the ability to fit the trajectory during blind time, an additional temporal average is introduced for sampling and non-sampling times. From a temporal perspective, the flow trajectory for a determined point should be robust and smooth between any sequences. We use temporally averaged root-mean-square error (TRMSE) to measure trajectory smoothness, for every candidate pixel **x** in the spatial domain, and the RMSE along the temporal dimension is computed to measure the influence of outliers. After that, the errors of all pixels are averaged.

$$TRMSE = \frac{1}{N_v} \sum_{\mathbf{x}} \sqrt{\frac{1}{T} \sum_{a=0}^{T-1} ||F(t_a) - D_{gt}(t_a)||^2}, \tag{10}$$

where $N_v$ is the number of valid points, $\mathbf{x} \in \mathbb{R}^{N_s \times 2}$ is the coordinates of valid points. $T$ is the number of times with ground-truth displacements for each point, $t_a$ is the corresponding time.

## 5.3 Baseline

To generate the motion trajectory of each pixel, we choose the flow estimation method RAFT [12] and the point-tracking method PIPs [9] as our baselines. RAFT predicts the optical flow between consecutive frames, and these flows are linked by bilinear interpolation as mentioned in [9]. We use their pre-training model and fine-tune it on our dataset. PIPs is a recent pixel tracking algorithm with promising generalization performance. We retrain PIPs with 4 frames on our dataset and infer the corresponding movements of all pixels during whole input clips to ensure fairness of the comparison.

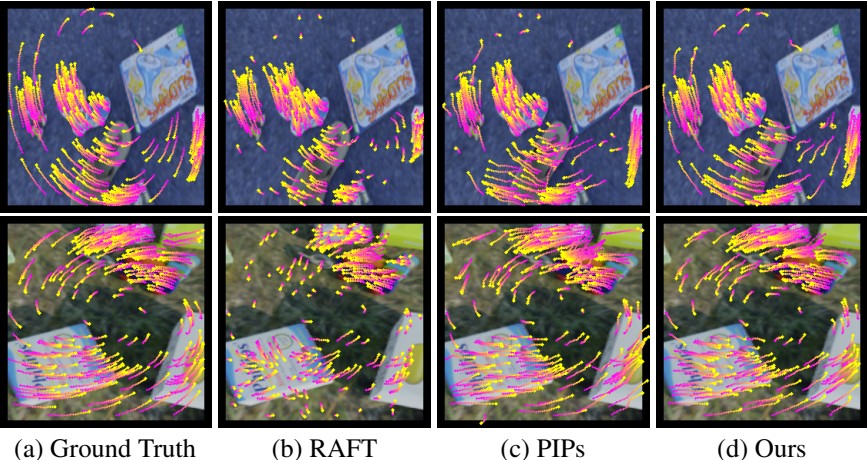

    (a) Ground Truth      (b) RAFT      (c) PIPs      (d) Ours

Figure 3: The visualization of dense point trajectories on the Query-First set.

Table 1: Evaluation on synthetic datasets. Our model with cubic B-splines and ODE encoder outperforms baselines on sparse validation (Query-Stride set) and dense test (Query-First set).

| Method | Metric | Mark | Query-Stride set | | | | Query-First set | | | |
|---|---|---|---|---|---|---|---|---|---|---|
| | | | 8f | 10f | 16f | 24f | 8f | 10f | 16f | 24f |
| RAFT | ADE_Vis | S | 8.18 | 7.91 | 13.00 | 14.43 | 5.68 | 7.18 | 11.12 | 17.56 |
| | | NS | 7.13 | 6.08 | 10.21 | 11.31 | 4.94 | 5.42 | 8.47 | 13.91 |
| | ADE_Occ | S | 9.62 | 12.16 | 27.73 | 24.95 | 6.26 | 8.60 | 14.77 | 24.86 |
| | | NS | 8.79 | 9.37 | 21.98 | 18.06 | 5.60 | 6.43 | 10.72 | 19.04 |
| | ADE_All | S | 8.32 | 8.17 | 14.93 | 16.37 | 5.73 | 7.27 | 11.44 | 18.49 |
| | | NS | 7.39 | 6.37 | 12.37 | 13.06 | 5.00 | 5.51 | 8.75 | 14.72 |
| | TRMSE | - | 8.50 | 7.82 | 15.81 | 16.66 | 5.86 | 6.77 | 10.35 | 16.85 |
| PIPs | ADE_Vis | S | 4.66 | 4.39 | 8.77 | 10.13 | 3.12 | 4.12 | 7.81 | 14.23 |
| | | NS | 3.97 | 3.40 | 6.92 | 7.92 | 2.97 | 3.26 | 5.97 | 10.27 |
| | ADE_Occ | S | 7.13 | 7.71 | 21.54 | 19.97 | 4.85 | 7.28 | 12.89 | 22.34 |
| | | NS | 6.43 | 5.93 | 16.83 | 15.16 | 4.23 | 4.03 | 9.31 | 16.35 |
| | ADE_All | S | 4.74 | 4.58 | 10.41 | 12.09 | 3.76 | 4.93 | 8.35 | 14.68 |
| | | NS | 4.13 | 3.61 | 8.68 | 9.86 | 3.25 | 3.76 | 6.42 | 11.76 |
| | TRMSE | - | 4.93 | 4.49 | **11.27** | 13.02 | 3.90 | 4.72 | 7.76 | 13.70 |
| ODE-6spline (Ours) | ADE_Vis | S | 3.81 | 3.88 | 8.08 | 8.22 | 2.88 | 3.72 | 6.26 | 12.08 |
| | | NS | 3.38 | 3.11 | 6.47 | 6.60 | 2.51 | 2.86 | 4.83 | 9.73 |
| | ADE_Occ | S | 6.24 | 8.09 | 22.72 | 19.84 | 4.70 | 6.86 | 11.34 | 19.61 |
| | | NS | 5.65 | 6.35 | 18.21 | 15.07 | 4.17 | 5.04 | 8.24 | 15.19 |
| | ADE_All | S | **4.00** | **4.13** | **9.89** | **10.61** | **2.96** | **3.88** | **6.71** | **13.05** |
| | | NS | **3.60** | **3.37** | **8.54** | **8.92** | **2.61** | **3.02** | **5.26** | **10.64** |
| | TRMSE | - | **4.26** | **4.17** | 11.38 | **12.30** | **3.12** | **3.78** | **6.42** | **12.38** |

\* S, NS are respectively referred to as the sampling and non-sampling moments.
\*\* f is an abbreviation of frames.

Since our goal is to predict continuous flow trajectories, not only corresponding to inputs, but also inter-frame trajectories. This allows us to set some of the whole frames in the dataset to be visible while others are hidden. The ground-truth displacements of the hidden frames can be used for the continuity evaluation, including inter-frame motion. Therefore, we introduce additional motion assumptions for the baselines to connect discrete points into complete continuous trajectories. We experimented with multiple assumptions, but the linear one yielded more stable results.

## 5.4  Evaluation on Simulated Data

We use two subsets of our synthetic dataset for validation in sampling ranges with 8, 10, 16, and 24 frames with uniform sampling. As mentioned earlier for the two evaluation perspectives, we evaluated inter-frame motion accuracy and multi-frame motion robustness, respectively. The result of the quantitative comparison is found in Table 1. As shown in Fig. 3, benefited by global optimization, our temporal parametric model outperforms baselines at all moments for all variable temporal ranges.

From the accuracy perspective, benefiting from continuous explicit modeling, our algorithm remains high performance at both sampling and non-sampling moments. The rate of error accumulation with the extension of the inference range is also controlled validly, especially for visible points. From the temporal stability perspective, compared with widespread polynomial motion approximation between reference and target frames, our curves based on limited variable control points can adjust the local trajectory flexibly for overall smoothness. Besides, our model is capable of adapting to both sparse and dense inference, particularly on dense estimation of short distances, which can sustain more consistent predictions. To reduce the instability of parametric modeling for a longer inference range, we expect to exert the powerful influence of implicit feature representation. By introducing the Neural ODE module with Conv-GRU update units, temporal features across multi-frames will be more convenient for motion separation and aggregation.

Table 2: Evaluation on real-world datasets. Our model with cubic B-splines and ODE encoder outperforms baselines on real scenes.

| Method | Metric | Mark | Vid-DAVIS [8] | | | | Vid-Kinetics [8] | | | |
|---|---|---|---|---|---|---|---|---|---|---|
| | | | 20f | 24f | 28f | 32f | 36f | 48f | 128f | 250f |
| RAFT | ADE_Vis | S | 16.84 | 19.72 | 19.42 | 23.37 | 15.97 | 19.00 | 30.19 | 35.93 |
| | | NS | 12.42 | 14.71 | 15.87 | 18.53 | 12.98 | 16.36 | 24.99 | 30.63 |
| | ADE_Occ | S | 18.58 | 23.82 | 27.91 | 31.35 | 19.23 | 21.89 | 31.98 | 39.76 |
| | | NS | 14.67 | 18.56 | 20.19 | 22.53 | 16.07 | 18.85 | 26.11 | 33.10 |
| | ADE_All | S | 17.99 | 20.25 | 22.97 | 25.65 | 16.74 | 20.70 | 31.47 | 37.81 |
| | | NS | 14.23 | 16.07 | 17.91 | 19.68 | 14.90 | 17.73 | 25.16 | 31.49 |
| | TRMSE | - | 16.91 | 18.95 | 21.28 | 23.81 | 16.62 | 24.51 | 27.50 | 34.23 |
| PIPs | ADE_Vis | S | 12.96 | 15.47 | 15.62 | 18.73 | 12.71 | 16.48 | 28.34 | 38.56 |
| | | NS | 10.39 | 11.39 | 12.60 | 15.27 | 10.49 | 13.30 | 22.31 | 31.08 |
| | ADE_Occ | S | 18.84 | 26.67 | 30.60 | 32.21 | 17.59 | 21.08 | 31.82 | 42.31 |
| | | NS | 14,42 | 20.66 | 20.88 | 23.26 | 15.42 | 18.56 | 25.34 | 35.41 |
| | ADE_All | S | 15.18 | 17.98 | 22.06 | 23.31 | 13.70 | 17.79 | 31.76 | 40.59 |
| | | NS | 11.94 | 14.38 | 16.62 | 18.49 | 12.73 | 13.72 | 24.67 | 33.90 |
| | TRMSE | - | 14.80 | 17.38 | 20.66 | 22.79 | 12.51 | 15.82 | 28.60 | 37.25 |
| ODE-6spline (Ours) | ADE_Vis | S | 11.37 | 16.80 | 16.16 | 18.99 | 12.09 | 15.02 | 25.25 | 31.18 |
| | | NS | 9.31 | 12.21 | 13.37 | 15.21 | 9.99 | 12.33 | 21.35 | 27.13 |
| | ADE_Occ | S | 15.75 | 20.67 | 26.71 | 27.99 | 17.76 | 20.50 | 29.79 | 37.12 |
| | | NS | 12.23 | 15.91 | 18.00 | 20.55 | 15.17 | 17.99 | 24.98 | 32.00 |
| | ADE_All | S | **13.48** | **16.91** | **19.81** | **21.16** | **12.47** | **15.82** | **27.67** | **34.42** |
| | | NS | **10.79** | **13.30** | **15.07** | **16.68** | **10.47** | **13.10** | **22.80** | **29.65** |
| | TRMSE | - | **12.96** | **15.76** | **18.12** | **20.27** | **12.06** | **14.91** | **25.44** | **32.78** |

## 5.5 Evaluation on Real Data

We choose the TAP-Vid-DAVIS and TAP-Vid-Kinetics datasets[8] to evaluate our model on real data, with 20,24,28,32 frames for evaluation on DAVIS and 36,48,128,250 frames including more non-sampling moments for Kinetics. For fairness of comparison, we need all methods to complete dense flow estimation and use identical interpolation techniques to obtain the corresponding trajectory of sparse points. The evaluation result is shown in Table 2 and the visualization result in Fig. 4.

According to the test result, we find our powerful model can surpass the flow-based method and point-tracking algorithm PIPs in all test ranges for all time ranges. Although PIPs perform well in specific point tracking, the model needs heavy costs for large numbers of points. We only infer 3072 points simultaneously for 256x256 resolution in the test stage with the 4-frame model. For dense validation, we have to process in batch with longer inference time while a flow-based solution needs more link steps resulting in more time consumption.

However, our model still shows potential in real scenes connected with our explicit continuous curve fitting. The motion trend is similar to ground truth even though there are slight trembles in the middle process. Thanks to large-scale training and implicit spatial-temporal continuous features, our work can keep relatively stable tracking which is helpful to reduce the drift in long-range prediction.

## 5.6 Ablation

We analyze two core components that contribute to our model on feature and trajectory continuity modeling. As for the feature encoder, the basic structure is from RAFT and no extra module is added for temporal representation. The ODE encoder will supplement an ODE unit with the combination of neural ODE function and Conv-GRU iterative part after basic feature extraction. As for the parametric curve, We set the hyper-parameter of the B-spline to reflect the number of control points which is designated 4 and 8. The result is shown in Table 3.

In the feature continuity aspect, ODE unit is valid for our parametric curve and equipped with a stronger ability to capture temporal-spatial connection. About the parametric curve selection, cubic B-spline with 4 and 8 control points are both weaker than 6 control points because less or more partition from control points will generate more loosen or tight constraints.

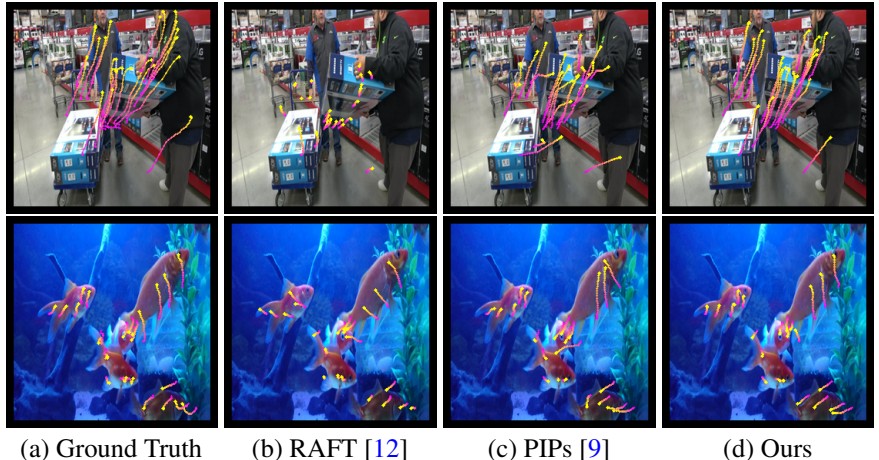

| (a) Ground Truth | (b) RAFT [12] | (c) PIPs [9] | (d) Ours |

Figure 4: The visualization of sparse point trajectories on Vid-DAVIS.

Table 3: The ablation of model components.

| Module | Method | Metric | 8f | 10f | 16f | 24f |
|---|---|---|---|---|---|---|
| Control Point | ODE + 4spline | ADE_All | 4.41 | 3.97 | 9.35 | 9.68 |
| | | TRMSE | 4.83 | 4.57 | 12.09 | 12.65 |
| | ODE + 8spline | ADE_All | 3.94 | 3.91 | 9.06 | 9.13 |
| | | TRMSE | 4.40 | 4.47 | 11.66 | 12.14 |
| Neural-ODE | Basic + 6pline | ADE_All | 4.89 | 4.43 | 9.79 | 9.92 |
| | | TRMSE | 5.45 | 5.04 | 12.41 | 12.93 |
| All | ODE + 6spline | ADE_All | 3.77 | 3.62 | 8.80 | 8.86 |
| | | TRMSE | 4.26 | 4.17 | 11.38 | 12.03 |

## 6 Conclusion

We have presented continuous parametric optical flow, a parametric representation of dense and continuous motion, which describes real pixel motion trajectory and expects to promote frame-to-frame discrete flow into continuous time-to-time motion. We generate a synthetic dataset based on Kubric for our task and introduce evaluation rules for continuity. With multi-frame inputs, a pipeline with the fusion of continuous features and the parametric curve is proposed for flow estimation. Experimental results on synthetic and real scenes reveal that our method focuses more on unseen inter-frame trajectory and there is room for development for all moments.

**Limitations and Broader Impacts.** As a parametric representation, the ability of our method to handle challenging complex motions will be limited by the choice of the degree of model and we have to admit the limitation in all parametric models, where a strong explicit constraint potentially leads to better performance on those more similar motion scenery with the training set. Our work is foundational research, for which we do not foresee it will bring negative social impacts. We also believe that the ethical concerns of our method in the training and evaluation process are minimal and there is no harm or bias to anyone. All weight parameters rely on learning from valid synthetic data [11] and real data utilized for evaluation comes from TAP-Vid [8] with a clear statement for the fairness and unbiasedness of these benchmarks.

**Acknowledgments.** This research was supported in part by the National Natural Science Foundation of China (62271410, 62001394), the Fundamental Research Funds for the Central Universities, and the Innovation Foundation for Doctor Dissertation of Northwestern Polytechnical University (CX2023013).

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
