# Continuous Parametric Optical Flow
# –Supplementary Material–

**Jianqin Luo**[*]    **Zhexiong Wan**[*]    **Yuxin Mao**    **Bo Li**    **Yuchao Dai**[†]

Northwestern Polytechnical University, Xi'an, China
Shaanxi Key Laboratory of Information Acquisition and Processing

## Abstract

In this supplementary material, we provide the details about our training dataset generation and analyse the occlusion distribution of point trajectory in real-world benchmarks. We also consider another explicit curve to compare with our model and provide extra visualization results of synthetic dataset. Besides, the error bars of our model is also reported.

## 1   Dataset for Training and Evaluation

In order to provide more information and explanation about our training data and evaluation benchmark, we introduce our synthetic samples with respect to the length and density of point trajectory. We also analyse the validity and occlusion situations of the real-world data that we used in our main paper.

### 1.1   Synthetic Dataset

We generate our synthetic data based on two different sampling strategies: Query-Stride and Query-First. The points in the Query-Stride subset are sampled from all frames and the initial frame varies for each query. The points in the Query-First subset are sampled from the first frame of a specific sequence. Moreover, the total number of sampled point trajectories for one clip is 65536 ($256 \times 256$) for both subsets, which means that the sampling points are relatively sparse in Query-Stride and almost dense in Query-First. In our experiment, we use point trajectories with various track lengths in Query-Stride to create a temporal augmentation in the training phase. We do not use completely dense supervision like ground truth in Query-First because it may introduce spatial redundancy and reduce the model's performance on long-term stability of single trajectory estimation. Therefore, we split the Query-Stride subset into training and validation sets and use the Query-First subset for dense testing.

We use the trajectories of query points in the Query-Stride subset to cover a diverse temporal scale. However, the sparse temporal-spatial sampling procedure for a given scene may slow down the training speed and hinder the spatial information aggregation for better dense prediction. To address this issue, we ensure that the points from the same scene start from the same frame and limit the variable-length sampling procedure to different scenes. We use the frame that contains most visible points as the reference and trim the rest of the clip to construct the training sequence. Figure 1(a) shows the scenes with different lengths that reflect the diversity of sequence lengths. We select four evaluation lengths based on the minimum, maximum and two other suitable lengths that cover more scenes. Moreover, as shown in Figure 1(b), the density of point trajectories on the reference frame is relatively high (83%-88%) but not fully dense. This is beneficial for dense inference in the validation stage and for spatial-temporal information fusion in the training stage. We also apply

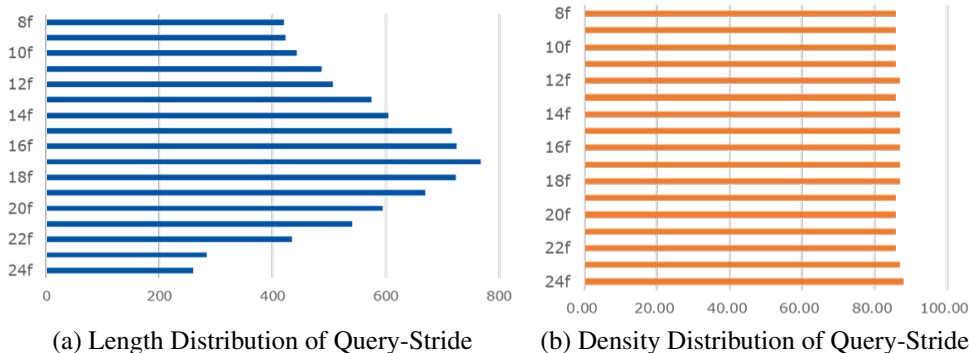

(a) Length Distribution of Query-Stride          (b) Density Distribution of Query-Stride

Figure 1: Statistics of Samples in Query-Stride subset

additional random spatial-sampling with a ratio of nearly 32% in the training stage to achieve a trade-off. Query-First mainly provides fully dense validation with 100 scenes for four inference lengths.

## 1.2 Real-World Benchmark

We use the TAP-VID-DAVIS and 999 scenes with 250 frames from TAP-VID-Kinetics as our main evaluation datasets. For each dataset, in the evaluation stage, we only use the visible points on the first frame for inference and the validity ratio is 83% and 75%, respectively. For all valid points, we define a visible query as a specific point that has complete visibility on all sampling moments. Otherwise, it is occluded. For VID-DAVIS, due to the minimum length of all scenes being 34, we select 20, 24, 28, 32 to cover the normal training range and short-term expansion with more non-sampling moments under higher temporal resolution. For VID-Kinetics, a longer inference range can provide a better approximation for continuous space. Therefore, we introduce 128 and 250 as evaluation scales. Moreover, as shown in Figure 2 and Figure 3, a longer inference length implies more frequent occlusion with great importance for testing the performance of the model on tracking pixels through occlusion in a real continuous environment.

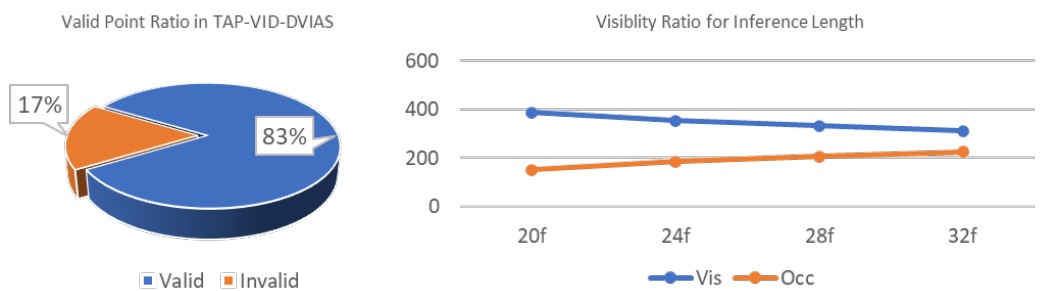

Figure 2: Point Validity and Occlusion in VID-DAVIS

## 2 Ablation Studies on the Curve Type

Similar to B-spline, the Bezier curve is also determined by a few control points and flexible to adjust the trajectory shape. Therefore, we choose the cubic Bezier curve as our explicit model and combine it with neural ODE module for training and evaluation. The comparison result is shown in Table 1. Cubic Bezier outperforms B-spline with the same degree only on 8-frame inference range and with the extension of inference length and more non-sampling moments, B-spline performs superior to Bezier. In contrast to the global optimization like Bezier, every control point of B-spline only determines a

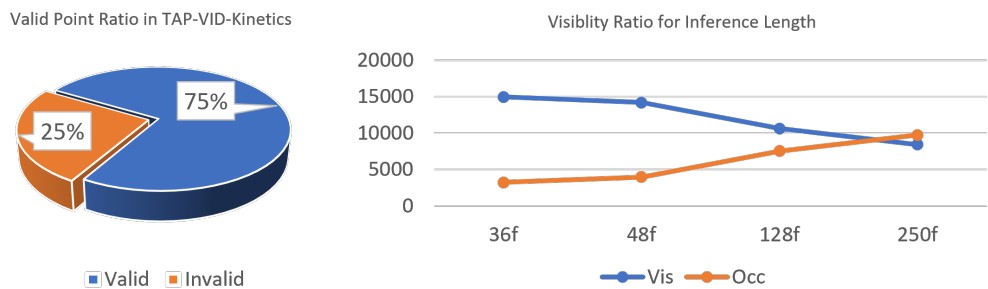

Figure 3: Point Validity and Occlusion in VID-Kinetics

part of the whole trajectory so fine-tuning is easier for B-spline to implement for adaptive trajectory adjustment.

Table 1: Performance comparison between Bezier and B-spline

| Method | Metric | Query-Stride-Val | | | |
|---|---|---|---|---|---|
| | | 8f | 10f | 16f | 24f |
| ODE + cubic-Bezier | ADE_all | **3.64** | 3.67 | 8.95 | 9.20 |
| | TRMSE | **4.10** | 4.24 | 11.52 | 12.39 |
| ODE + cubic-6spline | ADE_all | 3.77 | **3.62** | **8.80** | **8.86** |
| | TRMSE | 4.26 | **4.17** | **11.38** | **12.03** |

# 3   Error Bar

We run our training experiments three times to avoid the randomness of the experimental procedure from interfering the validity of conclusions. According to the result from validation set of Figure 4, our model is steady and robust to handle with the estimation of continuous parametric flow during different inference lengths.

Figure 4: Error bars of our model