# OpenReview forum: "Continuous Parametric Optical Flow"
_NeurIPS.cc/2023/Conference — NeurIPS 2023 poster_

### Official Review · Reviewer_TYqX · 2023-06-21

**Soundness:** 3 good
**Presentation:** 3 good
**Contribution:** 3 good
**Rating:** 5
**Confidence:** 2

**Summary:**

The submission 6736, entitled "Continuous Parametric Optical Flow," presents a novel multi-frame optical flow strategy that expresses the flow continuously. This is in contrast to conventional flow strategies, which encode this displacement in a discretized manner. This result is made possible by regressing the B-spline parameters of the pixel displacement. The approach shares similarities with RAFT and PIPs, in that this motion is refined iteratively via a convolutional-GRU, and the network can accommodate a series of successive images as input (more than 2). An extensive series of experiments underline the approach's effectiveness against existing state-of-the-art in optical flow and keypoint tracking. Additionally, novel datasets specifically dedicated to the task at hand have been created.

**Strengths:**

* The idea of estimating continuous optical flow from a neural network seems relatively novel.
* The performance reported in the manuscript is very competitive.

**Weaknesses:**

* While the overall editorial quality of the paper is acceptable, it still contains a large number of typos, and certain parts of the paper would benefit from additional proofreading. The flow of the paper sometimes lacks smoothness and transitions.
* The literature review is rather short and fails to position the paper with respect to other approaches. It gives the impression that the paper is poorly motivated. It would be helpful to underline more clearly the differences with other techniques and why this approach is substantially better.
* Similar to the previous comment, it would be interesting to list what downstream applications could benefit from this technique (3D? Tracking? Stabilization?).
* One of my main gripes regarding this manuscript is the lack of scalability. Due to its ability to input multiple frames, the network requires a significant amount of memory. This limits its applicability and raises the question of fairness when compared with less memory-intensive techniques.
* Another important point to address is the lack of clear contribution. In the reviewer's opinion, the main novelty of this paper is the type of output as B-splines, while the rest of the work is strongly inspired by other techniques.
* What are the scalability and clear limitations of the technique? How does it perform under very small or very large motions? It would be interesting to investigate such specific use cases.
* In Table 3, "Neural-ODE" is written, but in the method section, it is written as "Basic" while all other techniques are using ODE.


**Questions:**

I have expressed most of my questions in the previous section of this review. Despite all the shortcomings mentioned earlier, I found this work new and interesting, and therefore, I would like to issue a relatively positive opinion regarding its acceptance. However, the reviewer would like to specify that he has not been working on such a topic lately and has low confidence.


**Limitations:**

See previous sections.

---

> ### Author Rebuttal · Authors · 2023-08-09
>
> *1.Typos and Fluency:*
> - **Response:** Thanks for the valuable suggestion. We will fix all the typos and proofread the manuscript as suggested.
> ***
> *2.Motivation & Related Work:*
> - **Response:**
> - Thanks for the suggestion. In the paper, first, as agreed by all the reviewers, our novel concept of *continuous optical flow* describes **dense and continuous pixel motion displacement** regarded as an extension of classic optical flow dependent on adjacent frames. Inspired by RAFT's iterative optimization, we use similar techniques to construct cost volume and set the proper decoder for our task but introduce a continuous encoder and fuse explicit parametric curve. Notably, the step-by-step method based on chained flow is inevitable to generate drift and fails through occlusion. **In summary, our flow can generate all continuous correspondence at once with better accuracy after occlusion.**
> -  Secondly, as for the closest vision task, generic point tracking also achieve long-term fine-grained matching for frame-to-frame mapping.  With large-scale synthetic training and multi-frame aggregation, these methods like PIPs achieve better generalization ability and robustness. However, there are two drawbacks in current techniques: the discrete correspondence far from continuous displacements inherently and sparse tracking hard to infer in parallel for all pixels at once. **In summary, our method could generate dense and continuous point tracking more efficently**.
> - Thirdly, as for the continuous modeling, we investigate other motion estimation methods. In the video interpolation field, there are some classic motion assumptions like linear, quadratic, and cubic assumptions. For the balance of flexibility and accuracy, we choose B-spline with multiple adjusted control points as our regression objective which is rare in previous papers. **Utilizing a parametric but more flexible representation is also more suitable for continuous motion modeling.**
> - To conclude, continuous parametric optical flow is proposed to describe *spatially-dense and temporally-continuous pixel motion* with continuous encoded features and flexible parametric curve. **The current techniques are greatly limited for this task.**
> ***
> *3.Downstream Applications:*
> - **Response:** Thanks for the comment. Our method could provide spatial-temporal long-term correspondences for 3D vision tasks such as non-rigid structure-from-motion[1]  and simultaneous localization and mapping [2]. Another widespread downstream application is video analysis of dynamic scenes like arbitrary frame-rate video interpolation [3] and human keypoint tracking [4]. For the promotion of this field with more real motion data,[5] recently releases a novel large-scale dataset with high-quality annotation.
> ***
> *4.Scalability:*
> - **Response:** Thanks for the comment. On the scalability issue of our method, we think that multi-frame inputs require more memory than iteratively-update methods based on classic optical flow. But our method could support longer frame inputs with a slight increase in memory because we randomly select a fixed number of features to guide the model. Compared to those methods with less memory, our method will show better performance on accuracy and run time.
>
> Table Comparison of Runtime and Memory Cost with Differnet Inputs
> | Input Frames | Runtime/ms | Memory/MB |
> |:------------:|:----------:|:---------:|
> |       4      |    274.6   |    2442   |
> |       8      |    280.3   |    2456   |
> |      12      |    304.3   |    2482   |
> |      16      |    307.4   |    2503   |
>
> ***
> *5.Contribution:*
> - **Response:** Thanks for the comment. First, the novel concept of *continuous optical flow* is proposed to describe dense and continuous pixel motion with an explicit parametric curve B-spline.
> Secondly, The combination of neural optimization and neural ODEs (ODE-ConvGRU) for optical flow computation also makes novelty as agreed by the reviewers with different usage strategy and objective from existing video interpolation work. Thirdly, the field of continuous flow is almost blank, hence, the performance of algorithms needs a special evaluation system for nonsampling fitting and generalization. According to our investigation, it is also a valuable contribution to the training on an independent simulation dataset.
> ***
> *6.Limitations:*
> - **Response:** Thanks for the constructive comments. In terms of the scalability of our method, as mentioned in the above response, we need multi-frames as inputs with a slight increase in memory but obtain better performance for occlusion and inference speed. We think the limitation in our work is about the parametric model because strong motion prior will potentially obstacle learning for out-of-distribution generalization and long-term prediction because of the complexity of real motion.  As requested by the reviewer, here we provide some special cases like small or large motions in Figure 1 in the attached PDF.
> ***
> *7.Ablation:*
> - **Response:** Thanks for the comment. In ablation studies, the module ``Neural-ODE'' means we validate the function of this module. Hence,''Basic + 6spline'' denotes the setting that we use basic CNNs without the Neural-ODE module for evaluation while the full version with the Neural-ODE module is termed as ''All''. For other modules, the Neural-ODE encoder is used as the same setting to guarantee fair comparison.
> ***
>
> [1] Sidhu V, Tretschk E, Golyanik V, et al. Neural dense non-rigid structure from motion with latent space constraints.
>
> [2] Fu Q, Yu H, Wang X, et al. Fast ORB-SLAM without keypoint descriptors.
>
> [3] Park S, Kim K, Lee J, et al. Vid-ode: Continuous-time video generation with neural ordinary differential equation.
>
> [4] Kreiss S, Bertoni L, Alahi A. Openpifpaf: Composite fields for semantic keypoint detection and spatio-temporal association.
>
> [5] Zheng Y, Harley A W, Shen B, et al. PointOdyssey: A Large-Scale Synthetic Dataset for Long-Term Point Tracking.

---

> > ### Comment · Reviewer_TYqX · 2023-08-17
> >
> > I would like to thank the authors for all their clarifications and additional experiments they have conducted to address my concerns. Especially regarding limitation and scalability. I have also looked into other reviewer's comments and it comfort me in my first positive opinion regarding the acceptance of this paper for this conference.

---

> > > ### Author Response · Authors · 2023-08-17
> > >
> > > **Response**
> > >
> > > We would like to thank the reviewer for the positive and insightful comments. Regarding the limitations and scalability,  we are pleased to hear that the response is helpful in addressing the reviewer's concerns and look forward to receiving the acceptance notification for this conference.

---

### Official Review · Reviewer_JXsp · 2023-07-03

**Soundness:** 3 good
**Presentation:** 4 excellent
**Contribution:** 3 good
**Rating:** 5
**Confidence:** 4

**Summary:**

This paper proposed a continuous parametric optical flow estimation algorithm with B-spline temporal trajectory representation and ODE-ConvGRU based feature extraction. Experiment has been done on both synthetic and real-world dataset and the proposed continuous parametric method performs better than traditional flow-based and point-tracking based methods.

**Strengths:**

The idea of estimating continuous parametric flow is interesting and the achieved experimental result is promising. The presentation is clear and easy to follow. The ablation study clearly shows the benefits of B-spline representation and ODE-ConvGRU feature extraction.

**Weaknesses:**

1. It is somewhat unclear how to specify the flexibility of the trajectories, as in Table 3, different number of control points may affect the performance significantly, and in real scenarios, different objects in a scene may have different degree of motion complexities, it is not clear whether a single choice of B-spline number can handle this?
2. In table 2, the 'TRMSE' lines in 'PIPS' and 'ODE-6spline' under Vid-DAVIS dataset is exactly the same, this is problematic since their 'ADE_All' metrics are different, why?


**Questions:**

1. How about the computational complexity / running speed of the proposed algorithm? how about the memory cost?

**Limitations:**

I donot see any potential negative societal impact

---

> ### Author Rebuttal · Authors · 2023-08-09
>
> **Weakness**
> ***
> *1.Flexibility of Parametric Curve:*
> - **Comment:** It is somewhat unclear how to specify the flexibility of the trajectories, as in Table 3, different numbers of control points may affect the performance significantly, and in real scenarios, different objects in a scene may have different degrees of motion complexities, it is not clear whether a single choice of B-spline number can handle this?
> - **Response:** Thanks for the valuable comment. We agree that real-world  pixel motion could be rather complex. Consequently, for polynomial curves controlled by the only parameter *degree*, the flexibility will be limited.   In this paper, we choose B-spline as our fitting model because the shape of this curve is additionally dependent on adjustable control points regressed by our end-to-end framework. By adjusting certain control points, the local segment of the curve will be flexibly changed. Thus, for the splines, the degree only decides global complexity and the recovery of motion detail is achieved by locally optimization from multiple control points. Due to this partition optimization, **B-spline enables higher flexibility in handling complex motions**. Even though flexible spline curves like B-spline or Bezier curve could achieve improved ability in handling complex motions, we still have to admit the limitation in *all parametric models*, where a strong explicit constraint potentially leads to better performance on those more similar motion scenery with the training set.
> ***
> *2.Data Error:*
> - **Comment:** In table 2, the 'TRMSE' lines in 'PIPS' and 'ODE-6spline' under the Vid-DAVIS dataset are exactly the same, this is problematic since their 'ADE\_All' metrics are different, why?
> - **Response:** Thanks for pointing out this issue. We are sorry that we unexpectedly copy erroneous TRMSE results from raw data, displaying exactly the same performance in PIPs and ODE-6spline. Here we show the corrected tables as below.
> As the table reported, the TRMSE metric of ODE-6spline in fact outperform PIPs and RAFT in all inference range of the TAP-DAVIS benchmark and appear similar trend in other benchmarks.
>
> Table. Updated TRMSE metric of ODE-6spline in Vid-DAVIS benchmark
> |       Method       | Metric | Benchmark |    20f    |    24f    |    28f    |    32f    |
> |:------------------:|:------:|:---------:|:---------:|:---------:|:---------:|:---------:|
> | ODE-6spline (Ours) |  TRMSE | Vid-DAVIS | **12.96** | **15.76** | **18.12** | **20.27** |
>
> ***
> **Questions**
> ***
> *1.Computational Complexity & Memory Cost:*
> - **Comment:** How about the computational complexity / running speed of the proposed algorithm? how about the memory cost?
> - **Response1:** Thanks for the valuable comment. As requested, we provide the running speed and computational complexity below. Due to the samples in all benchmarks being 256x256, we use Vid-DAVIS to test the average runtime in the 20-frame inference range and memory cost for our method and all baselines.
> - **Response2:** Our method performs **more efficiently with shorter inference time** to generate spatially-dense and temporally-continuous pixel trajectories rather than baselines because an independent parametric curve for every pixel could directly provide all corresponding at once instead of multi-step iteration along temporal sequences based on original optical flow or along spatial coordinates based on long-term point tracker with sparse query points.
> - **Response3:** For faster inference, our proposed algorithm relies on multi-frame inputs and multi-time cost volume, resulting in a controllable increase in memory compared to traditional two-frame flow and robustness for tracking in parallel instead of significant expansion in memroy caused by single point tracker.
>
> Table. Comparison of Runtime and Memory Cost on Vid-DAVIS
> | Method | Runtime/ms | Memory/MB |
> |:------:|:----------:|:---------:|
> |  Ours  |    274.6   |    2442   |
> |  RAFT  |    338.2   |    1902   |
> |  PIPs  |   9073.7   |   18042   |

---

> > ### Comment · Reviewer_JXsp · 2023-08-18
> >
> > Thanks the authors for the detailed answers to my questions. All my concerns have been addressed and I don't have any further questions.

---

> > > ### Author Response · Authors · 2023-08-18
> > >
> > > We deeply appreciate the reviewer' valuable time and efforts in providing insightful comments on our submission. We are happy to hear that our detailed rebuttal has resolved all the questions.

---

### Official Review · Reviewer_sWYx · 2023-07-05

**Soundness:** 3 good
**Presentation:** 2 fair
**Contribution:** 3 good
**Rating:** 5
**Confidence:** 5

**Summary:**

This paper suggests a new model to pixel wise compute temporally continuous optical flow by using B-splines. The input to the neural model are sequences of images, the output are N 2D control points for each pixel of the spline model. During training, the input is sampled from the dataset at varying time instants and by varying number of samples. The model architecture builds on known design patterns such as CNN feature encoding, iterative optimisation (search) by using a recurrent GRU network, by using correlation pyramid to compute motion features. New elements are the ODE modelling of the continuous trajectories and the overall decoding (regression) of the control points for each pixel.

**Strengths:**

The paper introduces a model able to continuously capture motion in time. Applications like video editing could profit of such capabilities. The combination of neural optimisation and neural ODEs for optical flow computation seems novel to me.



**Weaknesses:**

Some insecurity in the argumentation, e.g. line 28: "real motion follows ... principles" - is there some motion of interest that does not follow physical principles?

The related work should be structured by the motion model, e.g. non-parametric motion, parametric motion. GMA is an attempt to overcome the limitations of non-parametric motion which is not clear by reading the text. The line of work called tracking (started with Lucas-Kanade) is non-parametric but sparse estimation of the apparent motion whereas particle video (Sand, Teller) tried to find a model in between dense optical flow and KLT.

line 160: xhi is not shown in Figure 2

line 166: computing correlation between features at reference time and target time is limited, e.g. think of in plane rotations.

There are typos in the text, e.g. line 254

line 233, Eq. 10: I do not understand "sampled point trajectories". Why is N_s needed in the metrics?

The spline is also continuous in the image, not only in time, but the ground truth available in the datasets  is discretly sampled, so how do you match an estimate F(t_k) to F* in Eq. 10?

**Questions:**

The model infers for all pixels in the reference image a 2NxWxH tensor with the control points of the pixel's B-splines. How do you infer the flow vector of all the pixels in the successive images (2nd, 3rd, ...) in the sequence?

Might trajectories of different pixels in the reference (1st) image collapse to a single pixel in some image in the sequence?



**Limitations:**

A first assessment of the impact has not been given by the authors. Even fundamental work on motion analysis & tracking should project potential impacts on foreseen applications and uses or policies such as the UN SDGs. Focus on computer vision as a field and try to explain how your research might affect the field in the next years. Try then to explain how the interaction of computer vision with society might be affected by your research, e.g. a seamlessly tracking combined with generative models might in future it even harder to distinguish fake video from the real.

Give real world examples why the B-spline model is limiting, why you need parametric motion and how you believe this dilemma could be overcome in future. The B-spline motion is smooth and introduces motion consistency which is in some cases a disadvantage, e.g. when it comes to abrupt motion cases.

---

> ### Author Rebuttal · Authors · 2023-08-09
>
> **Weakness**
> ***
> *1.Motion Claim:*
> - **Response:** Thanks for the comment. We agree that all the real-world motions follow classic physical principles. Thus, it is important to incorporate physical principles based on explicit constraints rather than directly using a neural network to regress point trajectories. Solely neural network based solutions may potentially cause abrupt changes that do not follow real-world physical principles.
> ***
> *2.Related Work:*
> - **Response:** Thanks for the valuable suggestion. Firstly, in this paper, we mainly highlight continuous pixel motion and propose the novel concept *Continuous optical flow* to describe and utilize the combination of parametric curve and implicit continuous feature to achieve flow estimation. Hence, the parametric or non-parametric modelling is just one part of the complete framework.
> -   In terms of the structure of related work, we divide into three subsets according to the relations with the concept, vision application using recent techniques and parametric motion assumptions. Every part is connected with different aspect.
> -  Besides, we do not state that GMA (Optical flow estimation) intend to overcome limitations of non-parametric motion and only cite it as an example of iterative update pipeline. For the tracking part, we intend to state some long-term point trackers could achieve stable frame-to-frame mapping with multi-frame inputs (exactly the whole sequence) and inference whole trajectory at once, which is similar with the effect of our work. As for the Particle Video, we intend to cite it as the inspiration of PIPs.
> ***
> *3.Figure Improvement:*
> - **Response:** Thanks for the comment. As suggested, we have updated Figure 2 as in the attached PDF.
> ***
> *4.Correlation Limitation:*
> - **Response:** Thanks for the comment. For in-plane rotations, correlation between features at reference time and target time based on matching framework can describe well without the influence of perspective effect. Thus, we do not believe there is problem in computing correlation there.
> ***
> *5.Typos:*
> - **Response:** Thanks for pointing out the typos. We will fix all the typos in the revised version.
> ***
> *6.TRMSE Explanation:*
> - **Response:** Thanks for pointing out the issue. Eq.~(10) aims to measure the trajectory smoothness by using the TRMSE metric. $N_s$ is the number of all valid pixel trajectories (sampled point trajectories) in spatial dimension. For every pixel, the algorithm will infer a continuous trajectory. In this metric, we expect to obtain a spatially-mean value through this parameter.
> ***
> *7.Contiguity Explanation:*
> - **Response:** Thanks for the comment. According to classic definition, optical flow is naturally dependent on 2D pixel coordinates. The parametric continuous curve in our work is only used for temporal representation. Hence, our model will provide an independent point trajectory for every pixel as mentioned in above TRMSE explanation. Considering this situation, we can directly sample corresponding ground truth at given moment for error measurement. In fact, the parametric model have been constructed when all control points are decided. Thus, for an arbitrary timestamp $t_k$, we could sample from this continuous curve and obtain the estimate $F(t_k)$.
> ***
> **Questions**
> ***
> *1.Trajectory Inference:*
> - **Response:** Thanks for the comment. As mentioned in our paper, the model generates all control points of B-splines for every pixel at once. According to the B-spline formulation, we only need a timestamp and could get corresponding flow vector relative to the reference time. Then, the flow vector of all the pixels in the successive images (2nd, 3rd, ...) in the sequence will be generated correspondingly. In our setting, the sequence of frames will be allocated a normalized timestamp calculated by division of frame index and sequence length. Besides, our model will directly infer long-range correspondences and not dependent on step-by-step iteration like classic optical flow limited to adjacent frames.
>
> *2.Pixel Independence:*
> - **Response:**  Thanks for the comment. In our paper, every single pixel's continuous flow is computed *independently*. Thus, whether multiple different trajectories collapse to a single pixel in some image does not matter any all. Interestingly, this will happen why occlusion is observed in 3D space.
> ***
> **Limitations**
> ***
> *1.Assessment of work:*
> - **Response:**  Thanks for the valuable comment.Through our method, stable continuous optical flow will provide realistic spatial-temporal corresponding priors for vision tasks such as non-rigid structure-from-motion, simultaneous localization and mapping, and video understanding of dynamic scenes. On the potential social impact, as the reviewer mentioned, continuous pixel motion trajectory could generate realistic dynamic change utilizing a few sequences and form a novel video. Hence, this technique can potentially make fake video with image generative framework and bring more challenges for official regulatory agencies. We will further elaborate the social impact as the reviewer suggested in details.
> ***
> *2.B-spline Limitations:*
> - **Response:**  Thanks for the valuable comment. We agree that real-world  pixel motion could be rather complex. Consequently, for polynomial curves controlled by the only parameter *degree*, the flexibility will be limited. In this paper, we choose B-spline as our fitting model because the shape of this curve is additionally dependent on adjustable control points regressed by our end-to-end framework. By adjusting certain control points, the local segment of the curve will be be changed with higher flexibility. Even though flexible spline curves like B-spline could achieve improved ability in handling complex motions, we still have to admit the limitation in *all parametric models*, where a strong explicit constraint potentially leads to better performance on those more similar motion scenery with the training set.

---

> > ### Comment · Reviewer_sWYx · 2023-08-14
> >
> > I appreciate the answers of the authors to my questions. After reading the other reviews and comments I decided to change my assessment. Using a parametrised motion model for OF is not new, but using a neural model of B-Splines is to a certain extend novel to the vision community. However, I have to admit that the limitations and the broader impact of this work needs to be elaborated in its own section in the paper. Following the conference code of conduct, ethical and societal impacts must be addressed adequately in the paper.

---

> > > ### Author Response · Authors · 2023-08-14
> > >
> > > **Response:**
> > >
> > > We would like to thank the reviewer for the prompt and insightful comments. We are glad to hear the reviewer decided to change the assessment.
> > >
> > > Here we would like to rephasize our main contributions. **First**, the novel concept of **continuous optical flow** is proposed to describe dense and continuous pixel motion with an explicit parametric B-spline curve. **Second**, the combination of neural optimization and neural ODEs (ODE-ConvGRU) for optical flow computation is also a novelty as agreed by the reviewers in solving the problem. **Third**, a specific evaluation system for this blank field is proposed with valid training based on an independent simulation dataset.
> > >
> > > In terms of the **limitations and broader ethical or social impacts**, we have mentioned them in the last part of our paper with relatively short statements and we will elaborate them in the revised version for a comprehensive discussion. As for the limitations, our model with parametric curves still faces challenges in capturing abrupt and complex displacements due to the trajectory smoothness and motion priors from the training datasets.
> > >
> > > We believe that the ethical concerns of our method, training and evaluation process are minimal and there is no harm or bias to anyone. All weight parameters rely on learning from valid synthetic data [1] with PII and real data utilizing for evaluation comes from TAP-Vid [2] with clear statement for the fairness and unbiasedness of these benchmarks. We also consider that the high-quality image generative model will potentially utilize continuous flow for more realistic video generation which could bring risks to the regulation of information source for social media. Thus, the implementation of whole algorithm will be authorized only for scientific purposes in the future.
> > >
> > > ***
> > > [1] Klaus Greff, Francois Belletti, Lucas Beyer, Carl Doersch, Yilun Du, Daniel Duckworth, David J Fleet, Dan Gnanapragasam, Florian Golemo, Charles Herrmann, et al. Kubric: A scalable dataset generator. In IEEE Conference on Computer Vision and Pattern Recognition (CVPR), pages 3749–3761, 2022.
> > >
> > > [2] Carl Doersch, Ankush Gupta, Larisa Markeeva, Adria Recasens Continente, Kucas Smaira, Yusuf Aytar, Joao Carreira, Andrew Zisserman, and Yi Yang. Tap-vid: A benchmark for tracking any point in a video. In NeurIPS Datasets Track, 2022, 35: 13610-13626.

---

### Official Review · Reviewer_dH6J · 2023-07-05

**Soundness:** 3 good
**Presentation:** 2 fair
**Contribution:** 2 fair
**Rating:** 5
**Confidence:** 5

**Summary:**

This paper presents a parametric representation of dense and continuous pixel motion over arbitrary time intervals. The  ``continuous parametric flow`` concept is interesting. However, one of the core technique contributions is encoding the image with the ODE-ConvGRU, which is not closely related to the  ``continuous parametric flow`` and simply replaces CNNs used in PIPs and RAFT.

**Strengths:**

1. The concept ``continuous parametric flow`` is interesting.
2. ODE-ConvGRU is a stronger image feature encoder compared with CNNs.
3. The B-spline-based flow interpretation is novel.

**Weaknesses:**

1. The comparison is unfair. The proposed model is trained with $N_{gt}=8$ but the PIPs is only trained with 4-frames. Can you provide the performance of an 8-frame PIPs model and the officially released PIPs model?
2. What does the ``implicit`` mean in the ``implicit feature representation``? For me, it's just a feature encoded by the ODE-ConvGRU instead of a CNN. I don't think it's necessary to introduce the ``implicit`` concept.
3. What's the relationship between the continuous flow with ODE-ConvGRU?
4. The two-stage evaluation is not novel, and should not be regarded as a contribution.
5. Figure 2 should be improved. The symbols in the ``cost volume`` and ``correlation pyramid`` regions are not well aligned. The legend of the concatenation is weird.

**Questions:**

See Weakness.

**Limitations:**

Limited by the chosen parametric model, the predicted point trajectory is challenging to express complex motions.

---

> ### Author Rebuttal · Authors · 2023-08-09
>
> **Weakness:**
> ***
> *1.Unfair Comparison:*
> - **Comment:** The comparison is unfair. The proposed model is trained with $ N_{gt} = 8 $ but the PIPs is only trained with 4-frames. Can you provide the performance of an 8-frame PIPs model and the officially released PIPs model?
> - **Response1:** Thanks for the comment. In terms of the training setting, we expect to utilize a few sampling frames as inputs and achieve trajectory estimation through continuous optical flow estimation. **All baselines including PIPs and our model use the same e.g. 4 sampling frames as inputs**. However, as the reviewer mentioned, the number of supervision moments $N_{gt}$ is a different concept from input frames. According to our assumption,**all moments in certain videos are valid for our continuous model.** Hence, the supervision can be randomly selected from these sampling or un-sampling timestamps with real spatial-temporal corresponding. But for the discrete point-tracking methods, the supervision is limited to selected from sampling moments, which causes **the supervision number is equal to inputs**.
> - **Response2:** As the reviewer mentioned, the official implementation of PIPs is an 8-frame inputs model, we will use this PIP model for comparison with our method with 8-frame inputs. **The performance on real-world benchmarks is shown in Table 1 in the attached PDF**. As the table reports, our 8-frame model still outperforms  PIPs with the same inputs in almost all inference ranges. Even though more inputs will provide more context information and improve accuracy and robustness at sampling moments for both methods, our proposed continuous parametric flow seems more stabel and smooth at non-sampling moments especially during the long-term blind time and maintain tracking  process through occlusion.
> ***
> *2.Implicit Representation:*
> - **Comment:** What does the implicit mean in the implicit feature representation? For me, it's just a feature encoded by the ODE-ConvGRU instead of a CNN. I don't think it's necessary to introduce the implicit concept.
> - **Response:** Thanks for the comment. In our paper, the *implicit* concept is used as the counterpart to explicit parametric constraints. The implicit features generated by ODE-ConvGRU aim to aggregate continuous spatial-temporal information and this could be understood as a kind of *memory* for a continuous video sequence similar to the video compression work Nerv[1].
> ***
> *3.Relations between Continuous Flow and ODE-ConvGRU:*
> - **Comment:** What's the relationship between the continuous flow with ODE-ConvGRU?
> - **Response1:** Thanks for the comment. We employ the ODE-ConvGRU unit to generate temporally continuous features defined at arbitrary timestamps by utilizing a handful of image embedding at sampling moments. These features will be used with random sampling for the construction of multi-time cost volume pyramids, which eventually guide the regression of control points in an iterative manner. Through these control points, every flow trajectory will be represented with an explicit B-spline curve.
> - **Response2:** The key part is how to obtain spatial-temporal features at *all moments* by limited temporal-free embedding at sampling moments. The first step is to aggregate all sampling features (the abbreviation of spatial embedding at sampling moments) for a suitable initial value in solving an ordinary differential equation (ODE). ConvGRU is used for this purpose. Neural ODE aims to compute feature differential by independent neural network and finally obtain continuous features by adaptive built-in ODE solvers like Euler or Runge-Kutta method.
> ***
> *4.Two-Stage Evaluation:*
> - **Comment:** The two-stage evaluation is not novel, and should not be regarded as a contribution.
> - **Response:** As stated in our contribution, the two-stage evaluation pipeline is designed for continuous flow estimation as there is no related work for this area. In fact, we hope continuous flow should equip with two attributes: one is the fitting ability at non-sampling or blind moments, and the other is for arbitrary time-to-time correspondence. Considering that the candidate moments for the continuous task are unlimited, we have to validate our method by the above two sub-tasks. We believe this also makes a contribution to the community.
> ***
> *5.Present Improvement:*
> - **Comment:** Figure 2 should be improved. The symbols in the cost volume and correlation pyramid regions are not well aligned. The legend of the concatenation is weird.
> - **Response:** Thanks for the suggestion. We will modify the legends of Figure 2 and ensure the aesthetic of the whole figure. The revised Figure 2 is provided in the attached PDF.
> ***
> **Limitations:**
> ***
> *1.Parametric Model:*
> - **Comment:** Limited by the chosen parametric model, the predicted point trajectory is challenging to express complex motions.
> - **Response:** Thanks for the comment. We agree that all parametric models depend on motion prior in special scenes and appear limited to real-world complex motions. However, different from polynomial curves controlled by the only parameter *degree*, in this paper, we choose B-spline as our fitting model because the shape of this curve is additionally dependent on adjustable control points regressed by our end-to-end framework and partition knots. By adjusting certain control points, the local segment of the curve will be flexibly changed. Due to this partition optimization, B-spline enables higher flexibility in handling complex motions.
> ***
> [1] Chen H, He B, Wang H, et al. Nerv: Neural representations for videos[J]. Advances in Neural Information Processing Systems, 2021, 34: 21557-21568.

---

### Official Review · Reviewer_7yvr · 2023-07-05

**Soundness:** 3 good
**Presentation:** 2 fair
**Contribution:** 2 fair
**Rating:** 3
**Confidence:** 3

**Summary:**

A temporally continuous parametric optical flow method, based on B-splines, is presented. The proposed network takes as input L frames (and timestamps) and outputs a tensor of size 2NxHxW, i.e N control points for each pixel. In practice N=6. The network architecture relies on neural ODE, ConvGRU and multi-time correlation pyramid. The proposed method is evaluated against RAFT and PIPs and a synthetic dataset (which is a contribution of this paper) and two real datasets (Vid-DAVIS and Vid-Kinetics). An ablation study is also presented.

**Strengths:**

1. The idea of using B-splines to model a temporally continuous parametric optical flow, is simple and efficient.
2. The proposed architecture is not trivial and well thought out.
3. The method outperforms both RAFT and PIPs.

**Weaknesses:**

1. The paper contains many typos, e.g. l.152 feed - > fed, l. 154 nerual, l. 251 will, etc.).
2. Some results are strange, for instance, in Table 2, PIPs and ODE-6spline have the exact same TRMSE on Vid-DAVIS.
3. In Table 1, PIPs outperforms ODE-6spline in terms of TRMSE at 16f on the Query-Stride set (11.27 vs 11.38) but it is not in bold.
4. Using ODE-ConvGRU does not seem new (it is said it was proposed in [49]).

**Questions:**

Please answer to the above "weaknesses". I am currently recommending to reject the paper as I feel the contribution is rather limited for NeurIPS.

**Limitations:**

.

---

> ### Author Rebuttal · Authors · 2023-08-09
>
> **Weakness**
> ***
> *1.Typos*：
> - **Comment:** The paper contains many typos, e.g. l.152 feed - > fed, l.154 neural, l.251 will, etc.
>
> - **Response:** Thanks for pointing out the typos. We will fix all the typos and further proofread the manuscript.
> ***
> *2.Data Error*：
> - **Comment:** Some results are strange, for instance, in Table 2, PIPs and ODE-6spline have the exact same TRMSE on Vid-DAVIS.
> - **Response:** Thanks for pointing out this issue. We are sorry that we unexpectedly copy erroneous TRMSE results from raw data, displaying exactly the same performance in PIPs and ODE-6spline. Here we show the corrected tables as below.
> As the table reported, the TRMSE metric of ODE-6spline in fact outperform PIPs and RAFT in all inference range of the TAP-DAVIS benchmark and appear similar trend in other benchmarks.
>
> Table. Updated TRMSE metric of ODE-6spline in Vid-DAVIS benchmark
>
> |       Method       | Metric | Benchmark |    20f    |    24f    |    28f    |    32f    |
> |:------------------:|:------:|:---------:|:---------:|:---------:|:---------:|:---------:|
> | ODE-6spline (Ours) |  TRMSE | Vid-DAVIS | **12.96** | **15.76** | **18.12** | **20.27** |
> ---
> *3.Omission Mark*：
> - **Comment:** In Table 1, PIPs outperforms ODE-6spline in terms of TRMSE at 16f on the Query-Stride set (11.27 vs 11.38) but it is not in bold.
> - **Response:** Thanks for pointing out the issue. We agree with the reviewer that PIPs slightly outperform ODE-6spline in terms of TRMSE metric at 16f on Query-Stride set in Table 1. We will update Table 1 correspondingly as suggested. Note that our method still shows comparable performance at 16f and better temporal smoothness in other inference scales.
> ---
> *4.Contribution Clarification*：
> - **Comment:** Using ODE-ConvGRU does not seem new (it is said it was proposed in [49]).
> - **Response:** Thanks for the comments. Here would like to further clarify the main contributions of our work (as been stated in the main manuscript).
>
>      In the paper, we present *continuous parametric optical flow*, which is a parametric representation of dense and continuous motion over arbitrary time intervals. This contribution has been acknowledged by all the reviewers (Reviewer dH6J ``the concept of continuous parametric flow is interesting``, Reviewer sWYx ``seems novel``, Reviewer JXsp ``the idea of estimating continuous parametric flow is interesting``, and Reviewer TYqX ``the idea of estimating continuous optical flow from a neural network seems relatively novel``).Note that Reviewer 7yvr also marked our idea in developing continuous parametric optical flow as ``simple and efficient `` and ``the proposed architecture is not trivial and well thought out ``.
>
>      In solving the problem, we utilize ODE convolutional GRU (ODE-ConvGRU) to encode implicit continuous features and establish multiple cost volumes for refinement. This combination of neural optimization and neural ODES for optical flow computation also makes novelty as agreed by the reviewers.
>
>      In this paper, we employed the ODE-ConvGRU unit for our novel problem and solution framework while we have not claimed the architecture design of ODE-ConvGRU as our contribution. Note that, the usage strategy and objective of ODE-ConvGRU in this paper also differs from existing work such as Reference [49]. In our paper, the ODE-ConvGRU encoder will generate continuous spatial-temporal features defined at specific timestamps rather than discrete frames from a few visible inputs and provide targets for multi-time cost volume construction.

---

> > ### Comment · Reviewer_7yvr · 2023-08-18
> > **Response**
> >
> > Hello,
> >
> > Thank you for your answers.
> >
> > Even if the paper is technically sound, I still find the contributions limited for a conference like NeurIPS. Nevertheless, I can see that I am the only one recommending to reject the paper. As a consequence, I will keep my initial rating but I will not fight against the other reviewers if they keep recommending to accept the paper.
> >
> > Best regards,
> > 7yvr

---

> > > ### Author Response · Authors · 2023-08-19
> > > **Detailed comments**
> > >
> > > We thank Reviewer 7yvr’s efforts in the reviewing process.
> > >
> > > Thanks for acknowledging the our contributions as “technically sound”,“The idea of using B-splines to model a temporally continuous parametric optical flow, is simple and efficient”, “The proposed architecture is not trivial and well thought out” and “The method outperforms both RAFT and PIPs.”.
> > >
> > > In the rebuttal, we have addressed all the comments with clear **explanation**, **numerical analysis** and **illustration**.
> > >
> > > If the reviewer still have **any** detailed concern with respect to our submission, please raise any comment and we are happy to discuss.
> > >
> > > The Authors

---

### Author Rebuttal · Authors · 2023-08-10

**Global Response**

In the attached PDF, we provide two figures and one table. Table 1 reports the comparison with PIPs with 8-frame inputs on real-world datasets. Figure 1 shows Special Cases of extremely Small Motion & Large Motion. Figure 2 illustrated the updated pipeline of our proposed framework.

---

### Author Response · Authors · 2023-08-21
**More Discussions**

Dear PCs, ACs and Reviewers,

We would like thank you for your dedicated time and thoughtful efforts in reviewing our paper. Your feedbacks and insights have been incredibly valuable in shaping the direction and quality of our work.

As the discussion period draws to a close, we wish to express our sincere appreciations to those reviewers who have engaged in the discussion with us. Your input has greatly helped us improve our manuscript.

However, we also recognize that the busy schedules and commitments of some reviewers might have led to discussions not yet taking place. We kindly invite all reviewers, including those who have not yet engaged in discussions, to take this final opportunity for further discussion. We believe that your perspectives, expertise, and suggestions are essential in guiding our paper towards its fullest potential.
We understand that time is of the essence, and we greatly appreciate your consideration in allocating some time for this discussion before the deadline.

Thank you all once again for your dedication to the field and for the invaluable role you play. We look forward to the possibility of engaging in insightful discussions and further improving our work based on your esteemed input.

Best regards,

The Authors

---

### Decision · Program_Chairs · 2023-09-21

**Decision:**

Accept (poster)

**Comment:**

Four reviewers recommend "borderline accept". From their comments it transpires that they all have a rather positive view of the paper,  clearly acknowledging that there is a novel and interesting contribution. On the contrary, one reviewer recommends rejection, but actually the strengths they list in their review appear to outweigh the listed weaknesses. The AC has read the paper and the authors' rebuttal, and concluded that the combination of a spline-based parametric flow model with the NODE idea is novel enough to warrant acceptance.